# Perception towards cardiovascular diseases preventive practices among bank workers in Hossana town using the health belief model

**Lemlem Kifleyesus Amdemariam[1], Aregash Mecha Watumo[1], Epfrem Lejore Sibamo[2], Feleke Doyore Agide[1]***

1 School of Public Health, College of Medicine and Health Sciences, Wachemo University, Hossana, Ethiopia, 2 School of Public Health, College of Medicine and Health Science, Hawassa University, Hawassa, Ethiopia

* feledoag@yahoo.com

**Data Availability Statement:** All relevant data are within the manuscript and its Supporting Information files.

## Abstract

### Background

Cardiovascular diseases (CVD) are becoming a public health problem in Ethiopia, especially among those who have limited physical activity. Although bank workers are at an increased risk of contracting CVD, their participation in CVD preventive activities is not studied in Ethiopia. Therefore, this study aimed to assess the perception of bank workers towards CVD preventive behaviors and associated factors in Hossana town.

### Methods

A cross-sectional study was conducted on a sample of 258 participants from February 11 to 30/2020. A simple random sampling method was used to select study participants from the enumerated list of staff. Data was collected using a self-administered structured questionnaire and the collected data was entered and analyzed using SPSS version 20 software. Descriptive statistics and logistic regression analysis were done. A p-value less than 0.05 with 95% CI was considered to declare an association between independent and dependent variables.

### Results

A total of 253 respondents with response rate of 98.0% were participated. The study revealed that the likelihood of performing CVD preventive behaviors is 62.0%. Moreover, the study found that bank workers' exposure to passive smoking [AOR = 0.5; 95% CI: 0.23–0.98], level of alcohol consumed [AOR = 0.5; 95% CI: 0.01–0.54], regularly consuming fruit and vegetable in daily meal [AOR = 0.16; 95% CI: 0.03–0.80], perceived severity[AOR = 0.1;95% CI: 0.01–0.68], and cues to take action [AOR = 0.12;95% CI: 0.02–0.73] were identified as predictors of perception to engage in CVD preventive behaviors.

### Conclusion

The level of bank workers' perception of engaging in CVD preventive behavior was in a considerable state to design and implement intervention strategies. Behavior change

**Funding:** The author(s) received no specific funding for this work.

**Competing interests:** The authors have declared that no competing interests exist.

communication should be strengthened to improve their knowledge and perception of the severity of CVD and barriers so as to improve the likelihood of taking action.

## Introduction

Globally, cardiovascular disease (CVDs) is becoming a priority public health problem [1, 2]. Cardiovascular disease (CVD) refers to a group of disorders of the heart and blood vessels, including coronary heart disease (CHD), hypertension, cerebrovascular disease, peripheral artery disease, heart failure, rheumatic heart disease, congenital heart disease, deep vein thrombosis and pulmonary embolism [2, 3]. Nowadays, CVDs are recognized globally as the main cause of death and morbidity. In particular, in low-and middle-income countries, the number of people dying from cardiovascular diseases is increasing annually [3, 4].

According to the WHO global health estimate of 2016, cardiovascular diseases are responsible for 17.9 million deaths that occurred globally, accounting for 44% of NCD deaths and 31% of all global deaths [5]. As a result of this, Sub-Saharan Africa's epidemiologic transition has been coined as a 'double burden of disease', referring to the existence of both communicable diseases and non-communicable diseases (NCDs), where NCDs are projected to account for more than half of all deaths by 2030 in SSA [6, 7].

The American Heart Association (AHA) categorizes the risk factors associated with cardiovascular disease into modifiable risk factors such as smoking, dietary habits, abnormal alcohol consumption, physical activity, overweight and obesity, and non-modifiable factors such as age, gender, genetics and family history [8, 9].

In Ethiopia, according to the WHO report of 2018 on NCD, cardiovascular diseases were accountable for 16% of all deaths [9]. Evidence from the Global Burden of Disease Study indicated that the number of prevalent CVD cases in Ethiopia has shown a100% increase between 1990 and 2017. However, the burden of CVDs in Ethiopia might be even higher than anticipated because of the current epidemiologic transition as a result of demographic and lifestyle changes in the general population [10].

These behavioral risk factors may show up in individuals as raised blood pressure, raised blood glucose, raised blood lipids, and overweight and obesity and indicate an increased risk of developing a heart attack, stroke, heart failure and other complications [11]. CVDs like acute coronary events (heart attacks), hypertensive crisis and cerebrovascular events (strokes) result in gradual or sudden onset that is often fatal before medical care can be given [11, 12]. This requires taking proactive measures to prevent the occurrence of the disease, slowing down disease progression and responding to adverse cardiovascular conditions through primary, secondary and tertiary prevention mechanisms. The type of prevention intervention to be applied is decided by identifying where the individual is in the natural history of the course of cardiovascular disease [13]. Primary prevention of cardiovascular disease takes place before precursory signs of cardiovascular disease, prior to the onset of biological risk factors or at the pre-pathogenesis stage [14]. Secondary prevention requires taking measures at the initial stage of pathogenesis or disease occurrence [15]. Thus, reducing the chance of getting cardiovascular disease in those who are at high risk of cardiovascular disease and responding to clinical events early and minimizing premature death in people with established cardiovascular diseases at varying clinical points on their continuum is very important [16].

The best practices for the prevention and reduction of cardiovascular disease involve understanding respondents' risk perceptions and their engagement in protective behaviors. Since

bank workers spend most of their time working in offices, they are at a greater risk of contracting cardiovascular disease. Theories and models help in explaining how people engage in healthy behavior and react to different behavioral practices [17]. Hence, this study seeks to use the Health Belief Model (HBM) to assess bank workers' risk perceptions and practices about cardiovascular disease prevention. The Health Belief Model (HBM) is a socio-psychological model that attempts to explain and predict health behaviors in terms of certain belief patterns and by focusing on the attitudes and beliefs of individuals. It was developed in the 1950s as part of an effort by social psychologists in the United States Public Health Service to explain the lack of public participation in health screening and prevention programmes. Since then, it has been adapted to explore a variety of long and short-term health behaviors, including cardiovascular risk behaviors. The originators of the HBM conducted major studies that systematically explained preventive health behavior considering various perspectives, such as the world of the perceiver, health motivation, and the individual's current dynamics that can be influenced by prior experience as a determinant of what an individual will and will not do [18]. The HBM addresses the individual's perceptions of the threat posed by a health problem (susceptibility, severity), the benefits of avoiding the threat, and factors influencing the decision to act (barriers, cues to action, and self-efficacy). It states that perceptions of general health values, specific health beliefs related to the health problem and recommended health actions influence the likelihood of taking recommended health actions [17, 18].

Therefore, this study tried to assess bank workers' risk perceptions and practices about cardiovascular disease prevention using the HBM (**Fig 1**).

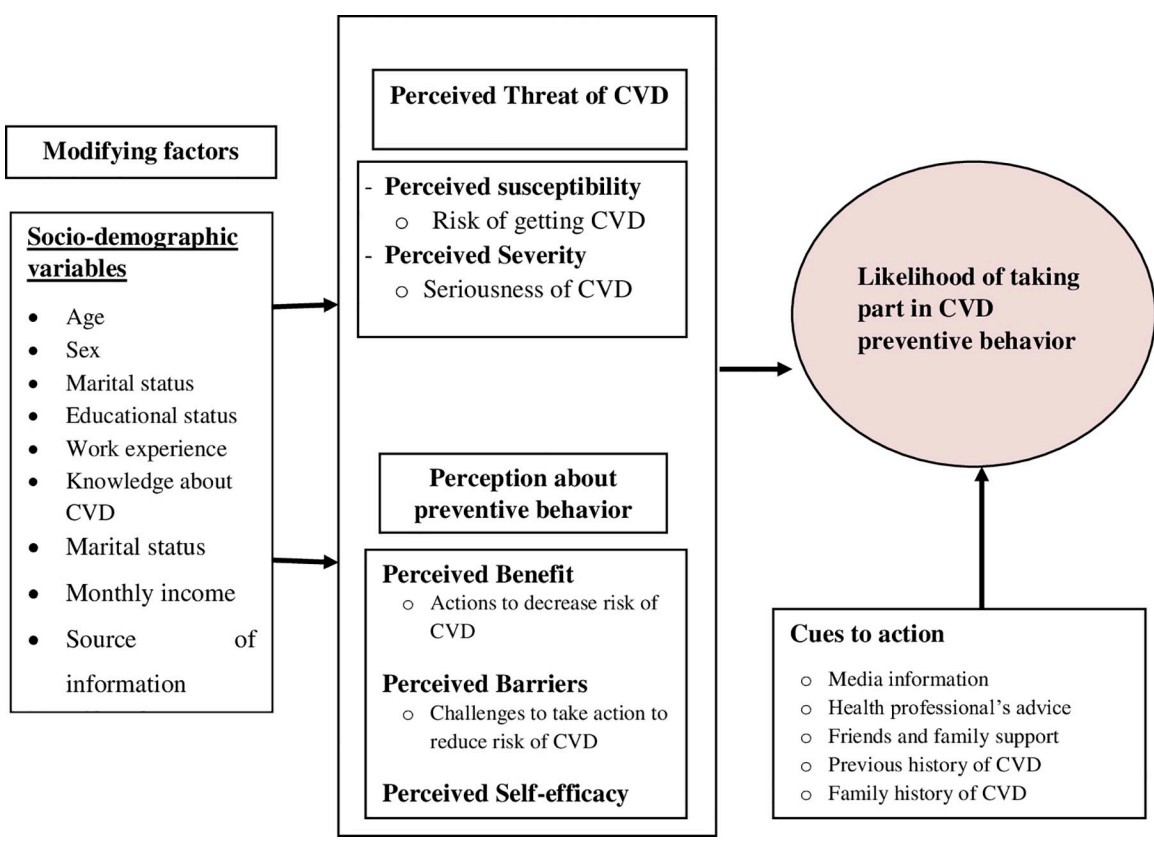

**Fig 1. Conceptual framework of the study.**

## Materials and methods

### Study area and period

This study was conducted in Hossana town. The town is located in the Hadiya Zone, being the capital of the Zone, which is located 232km from Addis Ababa, the capital of Ethiopia. In Hossana town, there are 16 banks with 39 branches (including both governmental and private Banks), with 681 workers. The study period was from February 11 to February 30, 2020.

### Study design and populations

A cross-sectional study was conducted among bank workers in Hossana Town.

### Sample size and sampling procedures

The sample size was determined by using a single population proportion formula considering the following parameters: since there is no study conducted on bank workers' perception of cardiovascular disease preventive behaviors, the sample size was calculated by assuming that 50% of the workers were engaged in cardiovascular disease preventive behaviors, marginal error (d) 5% and confidence interval of 95%. Then, the sample size will be 245. Finally, the sample size was further increased by 5% to account for contingencies such as non-response or recording errors, i.e. 245 X 5/100 + 245 = 257.5. Therefore, the final sample size was 258. A simple random sampling technique was used to select study participants from the enumerated lists of the staff records of each bank through a proportional to size (PS) allocation technique (Fig 2).

### Measurement and variables

The likelihood of taking preventive measures for CVD is the outcome of this study (perceived benefits minus perceived barriers). The exposure variables were socio-demographic factors, knowledge of CVD and its preventive measures, perceived susceptibility, perceived severity, self-efficacy, cues to actions, and past behaviors related to CVD preventive behaviors. Socio-demographic characteristics: such as age, sex, marital status, religion, education and occupation status and experience of the respondents. There are 7 knowledge questions with a response format of 'yes' or 'no'. Knowledgeable are those respondents who have answered 50% and above of all the knowledge questions about CVD prevention. Not knowledgeable were those respondents who could answer below 50% of all the knowledge questions about CVD prevention. Perceived susceptibility is the respondent's self-perception of vulnerability to CVD, measured by a summed score of related 4 belief items on a 5-point Likert scale. Perceived severity is the respondent's held belief concerning the effects of CVD seriousness, measured by a summed score of related 7 belief items on a 5-point Likert scale. Perceived benefits of CVD preventive measures is a respondent's belief about the effectiveness of the method as a strategy for CVD prevention, measured by a summed score of related 5 belief items on a 5-point Likert scale. Perceived barriers to preventing CVD are respondents' beliefs about the ease of performing the given preventive action by a summed score of related 9 belief items on a 5-point Likert scale. Self-efficacy is the respondent's confidence in using recommended preventive measures by himself/herself in any condition and elsewhere to prevent CVD, measured by a summed score of related 6 belief items on a 5-point Likert scale. Cues to actions are conditions that may facilitate them to perform preventive measures in the respondents' surroundings, measured by 4 items with a response format of 'yes' or 'no'. Behavioral risk factors/Past behaviors are any actions that are performed to prevent CVD in a lifetime measured with nominal measurements.

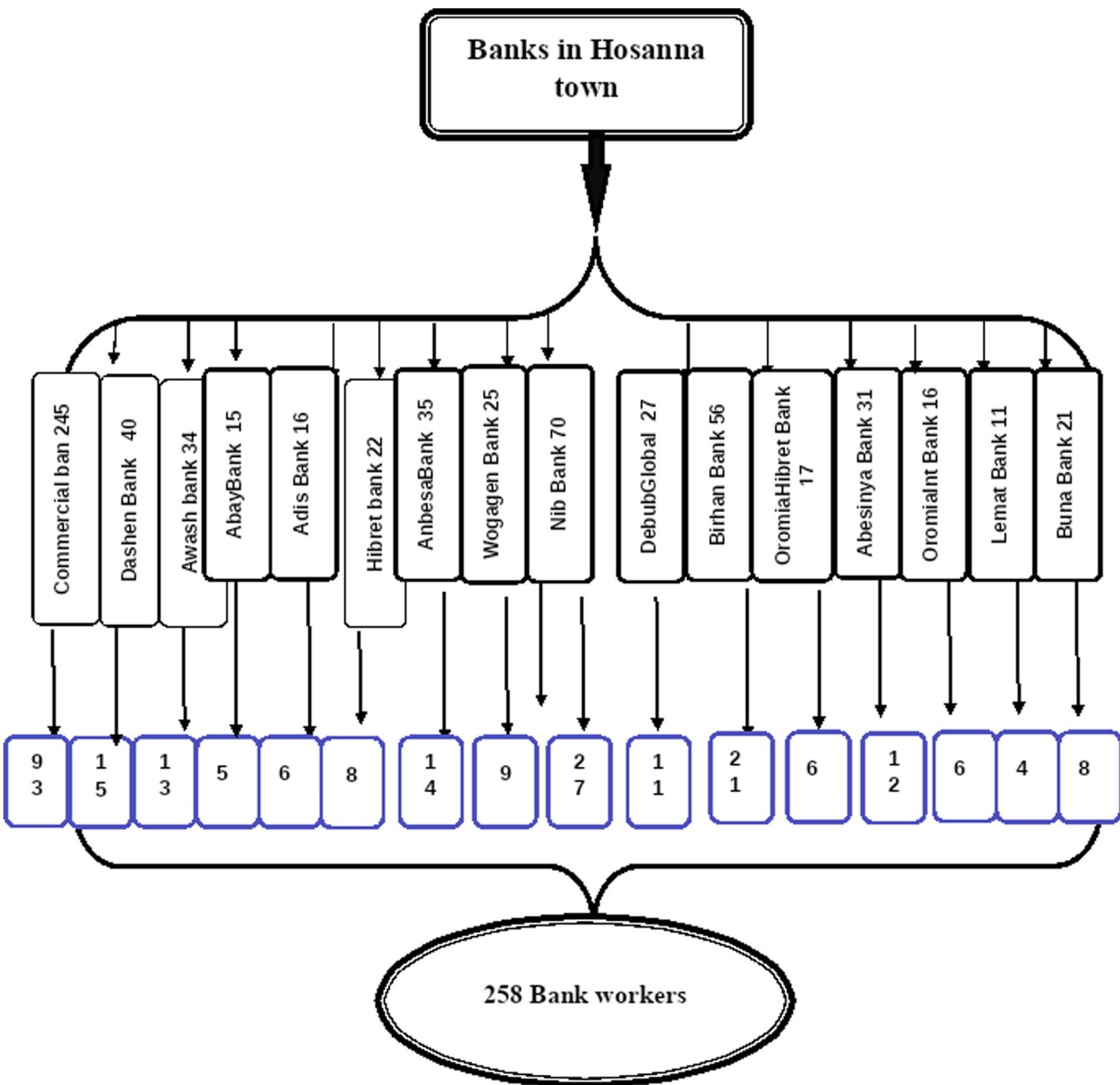

**Fig 2. Schematic presentation of sampling procedure to select study participants.**

Factor analysis was done for validation of the instrument. This was confirmed by considering a factor loading score of greater than or equal to 0.4 for construct validity. Cronbanch's Alpha was used to measure the internal consistency of items and was accepted when greater than or equal to 0.7. Accordingly, perceived susceptibility (4 items with Cronbach's alpha = 0.789), perceived severity (7 items with Cronbach's alpha = 0.745), perceived benefits (5 items with Cronbach's alpha = 0.898), perceived barriers (9 items with Cronbach's alpha = 0.750), perceived self-efficacy (6 items with Cronbach's alpha = 0.779), and cues to

action (4 items with Cronbach's alpha = 0.70). Each question was assessed by 5-point Likert scale responses, yielding a total score of 5–25 with Cronbach's alpha = 0.801.

## Data collection instrument and procedure

Data was collected using a self-administered structured questionnaire adapted from various studies conducted using the health belief model [19–24].

## Data quality management, processing and analysis

The questionnaires were primarily prepared into English and translated to Amharic, and then back-translated into English by another person to maintain consistency. The training was given to data collectors and supervisors. Investigators performed immediate supervision at the time of data collection. The data was analyzed by SPSS V. 20.0. For uniform scoring of items in the five-point Likert scale response format, negatively constructed items were reversed. Descriptive analysis was used to describe the percentages and number of distributions of the respondents by socio-demographic characteristics, knowledge and past behaviors and the main constructs of HBM. Furthermore, Binary logistic regression was used to identify the independent predictors of perception of CVD prevention. All explanatory variables that were associated with the outcome variable in bivariate analysis with a p-value of 0.20 or less were included in the initial logistic models to increase the candidate variables for final model prediction. The model fitness was measured by the *Hosmer-Lemeshow goodness* of *fit test* for logistic regression. The crude and adjusted odds ratios together with their corresponding 95% confidence intervals were computed and interpreted accordingly. A P-value of 0.05 and less was used to declare the variable statistically significant.

## Ethics

The study was conducted after securing ethical approval from Wachemo University as per the guidelines of the university. Permission was sought from the respective banks where the study participants were employed. Finally, after informing the participants about the purpose of the study, benefits and risks associated with the study, written consent was secured from each study participant before collecting the data. The participants were also informed that their responses would be kept confidential and their names would not be mentioned.

## Results

### Socio-demographic characteristics of the participants

A total of 253 respondents participated in this study, producing a total response rate of 98.0%. Table 1 presents the socio-demographic characteristics of the participants. Accordingly, the majority, 204 (80.6%), of the respondents were males. The mean (± SD) age of the respondents was 27.57 (+ 3.56) years (**Table 1**).

### Knowledge of respondents about CVD and its prevention methods

Figs 3 and 4 show the knowledge of the participants and the source of information for CVD preventive behavior. Accordingly, the comprehensive knowledge of the participants was 124 (56.6%) (**Fig 3**). And the majority, 233 (92.1%) of the study participants reported that they heard and knew about cardiovascular diseases (**Fig 4**).

**Table 1. Socio-demographic characteristics of bank workers in Hossana town, Ethiopia.**

| Variables | | Number | Percent |
|---|---|---|---|
| **Sex** | Males | 204 | 80.6 |
| | Females | 49 | 19.4 |
| **Age** | < 25 years | 84 | 33.2 |
| | 25–30 years | 123 | 48.6 |
| | > 30 years | 46 | 18.2 |
| **Education status** | Diploma | 28 | 11.1 |
| | Degree | 163 | 64.4 |
| | Masters | 62 | 24.5 |
| **Marital status** | Single | 128 | 50.6 |
| | Married | 116 | 45.8 |
| | Divorced | 2 | 0.8 |
| | Widowed | 7 | 2.8 |
| **Religion** | Orthodox | 65 | 25.7 |
| | Protestant | 137 | 54.2 |
| | Muslim | 32 | 12.6 |
| | Catholic | 19 | 7.5 |
| **Work position** | Beginner | 26 | 10.3 |
| | Officer position | 193 | 76.3 |
| | Mid-level manager | 19 | 7.5 |
| | Higher level manager | 15 | 5.9 |
| **Experience** | 1–3 years | 85 | 33.6 |
| | 4–6 years | 112 | 44.3 |
| | 7 years & above | 56 | 22.1 |
| **Weekly working hours** | 45 and less hours | 70 | 27.7 |
| | 46–50 | 124 | 49.0 |
| | Above 51 hours | 59 | 23.3 |
| **Monthly income** | < = 5000 Birr | 10 | 4.0 |
| | 5001–10000 Birr | 125 | 49.4 |
| | > = 10001 Birr | 118 | 46.6 |

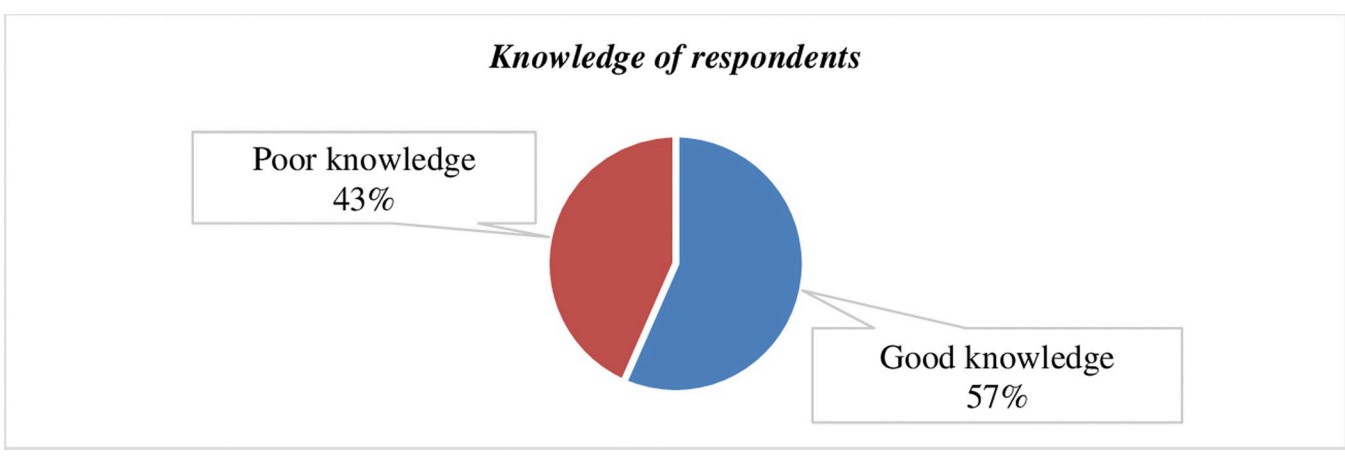

**Fig 3. Knowledge of bank workers about CVD among respondents in Hossana town, South Ethiopia.**

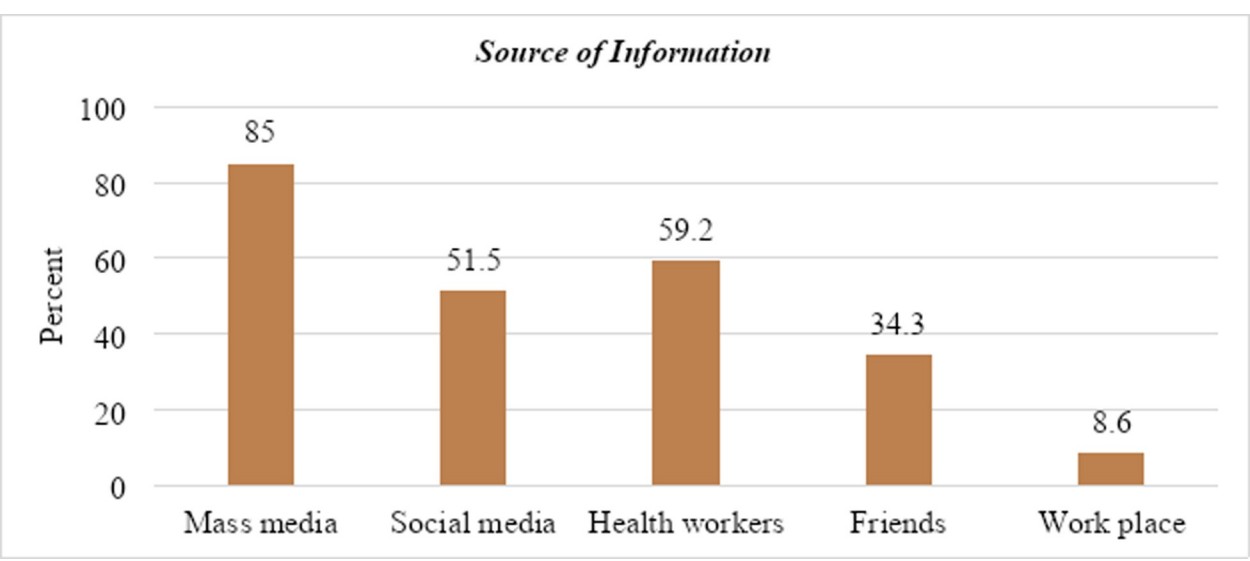

**Fig 4. Source information about CVD among bank workers in Hossana town, South Ethiopia.**

## Past behaviors and behavioral risk factors

Table 2 presents past behaviors and behavioral risk factors for CVD prevention among bank workers in Hossana town. Accordingly, the study revealed that only 13 (5.1%) of the participants had ever smoked cigarettes and 72 (28.5%) of them were exposed to passive smoking. The study also indicated that more than half, 144 (57.0%), of the respondents were currently engaged in regular physical activities (**Table 2**).

## Perception towards CVD and its preventive behavior

The mean score was calculated for each sub-scale to categorize respondents' level of risk perception. Table 3 presents the mean score of HBM constructs for preventive behaviors against

**Table 2. Past behaviors and behavioral risk factors related to CVD prevention among bank workers in Hossana town, South Ethiopia.**

| Characteristics | | Number | Percent |
|---|---|---|---|
| **Ever smoked cigarette** | Yes | 13 | 5.1 |
| | No | 240 | 94.9 |
| **Exposed to passive smoking** | Yes | 72 | 28.5 |
| | No | 181 | 71.5 |
| **Physical activity** | Yes | 144 | 56.9 |
| | No | 109 | 43.1 |
| **Alcohol taking** | Yes | 89 | 35.2 |
| | No | 164 | 64.8 |
| **Daily vegetable & fruit intake** | Yes | 94 | 37.2 |
| | No | 159 | 62.8 |
| **Daily excess salt & fat intake** | Yes | 218 | 86.2 |
| | No | 35 | 13.8 |
| **History of CVD** | Yes | 51 | 20.2 |
| | No | 202 | 79.8 |
| **Family history of CVD** | Yes | 79 | 31.2 |
| | No | 174 | 68.8 |

**Table 3. Mean score of HBM constructs for preventive behaviors against CVD among bank workers in Hossana town, South Ethiopia.**

| HBM constructs | Number of items | Mean ± SD | Range of scores |
|---|---|---|---|
| Perceived susceptibility | 4 | 11.24 ±3.42 | 4–20 |
| Perceived severity | 7 | 19.66 ± 4.95 | 7–35 |
| Perceived benefit | 5 | 19.61 ± 4.27 | 5–25 |
| Perceived barrier | 9 | 35.17 ± 5.69 | 9–45 |
| Perceived self-efficacy | 6 | 20.11 ± 4.10 | 6–30 |
| Cues to action | 4 | 12.41 ± 2.72 | 4–20 |

CVD among bank workers in Hossana town. Accordingly, perception of threat appraisals such as perceived susceptibility to and perceived severity of CVD had an average score of 147 (58.1%) and 137 (54.2%) respectively, whereas perceived benefits and barriers had an average score of 144 (57.0%) and 143 (56.5%) respectively (**Table 3**).

## Likelihood of taking preventive measures

The weighted mean score of the benefits is subtracted from the weighted mean scores of the barriers to yield the likelihood of taking part in CVD preventive behaviour of bank workers in Hossana Town. Fig 5 summarizes the perception of the participants' mean scores about the likelihood of taking part in CVD prevention. Accordingly, the likelihood of taking part in CVD preventive practice of bank workers in Hossana Town is 62.0% (**Fig 5**).

## The independent predictors of perception of CVD preventive behavior

Binary and multivariable logistic regression models were used to assess the effect of independent variables on the likelihood of participating in CVD prevention activities. Table 4 presents the independent predictors of CVD preventive behaviour. Accordingly, passive smoking, 5% alcohol drinking, regular consumption of fruit and vegetables, perceived severity and cues to action were significant crude and adjusted effects on the perception of CVD preventive behaviour. Passive smoking at home, workplace or other areas in the final model revealed that

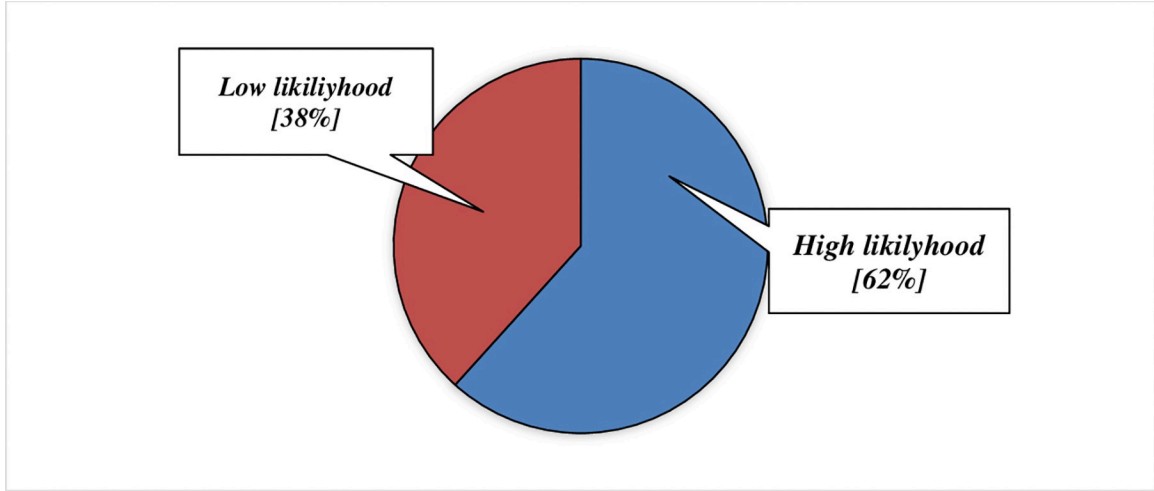

**Fig 5. Likelihood of CVD preventive behaviors among bank workers in Hossana town, Ethiopia.**

**Table 4. Logistic regression to identify factors associated with perception of CVD risk preventive behavior among bank workers in Hossana, South Ethiopia.**

| Variables | Perceived likelihood of taking action | | COR (95%CI) | AOR (95%CI) |
|---|---|---|---|---|
| | Lower No (%) | Higher No (%) | | |
| **Past smoking** | | | | |
| Yes | 36 (14.2%) | 36 (14.2%) | 0.51 (0.29, 0.88) * | 0.5 (0.23, 0.98) ** |
| No | 61 (24.1%) | 120 (47.4%) | 1 | 1 |
| **Alcohol consumption** | | | | |
| 5% alcohol | 25 (28.7%) | 42 (48.3%) | 0.39 (0.14, 1.10) | 0.06(0.01, 0.54)** |
| >5% alcohol | 12 (13.8%) | 8 (6.2%) | 1 | 1 |
| **Regular consumption of fruit and Vegetable** | | | | |
| Yes | 41 (16.2%) | 53 (20.9%) | 0.70 (0.41, 1.18) * | 0.16 (0.03, 0.80) ** |
| No | 56 (22.1%) | 103 (40.7%) | 1 | 1 |
| **Perceived Severity** | | | | |
| Low perceived severity | 60 (23.7%) | 56 (22.1%) | 0.34 (0.2, 0.58) * | 0.1 (0.01, 0.68) ** |
| High perceived Severity | 37 (14.6%) | 100 (39.5%) | 1 | 1 |
| **Perceived benefits** | | | | |
| Low perceived benefits | 57 (22.5%) | 52 (20.6%) | 0.35 (0.21, 0.59) * | 0.21 (0.06, 0.82)** |
| High perceived benefits | 40 (15.8%) | 104 (41.1%) | 1 | 1 |
| **Perceived barriers** | | | | |
| Low perceived barriers | 44 (26.2%) | 65 (39.5%) | 0.61 (0.34, 0.91) * | 0.53(1.13, 6.06)** |
| High perceived barriers | 21 (9.9%) | 123 (24.5%) | 1 | 1 |
| **Perceived Cues to action** | | | | |
| Low cues to action | 53 (20.9% | 66 (26.1%) | 0.61 (0.36, 1.01) * | 0.12 (0.02, 0.73) ** |
| High cues to action | 44 (17.4% | 90 (35.6%) | 1 | 1 |

* Variables significant at p–value < 0.20

** Variables significant at p–value < 0.05.

respondents who were exposed to passive smoking at home, workplace or other areas were 50% less likely to engage in CVD preventive practices compared to their counterparts, AOR = 0.5 [95% CI: 0.23–0.98]. Bank workers who used to drink 5% alcohol were 50% less likely to engage in CVD preventive behaviours compared to those who drank >5% alcohol, AOR = 0.5 [95% CI: 0.01–0.54]. The study also showed that respondents who regularly consume fruit and vegetables in their daily meal were 84% less likely to engage in CVD preventive activities compared to non-consumers of fruits and vegetables in their daily meal, AOR = 0.16 [95% CI: 0.03–0.80]. The perceived likelihood of engaging in CVD preventive activities among respondents with a history of CVD was 86% lower than those without a previous history of CVD, AOR = 0.14 [95% CI: 0.04–1.07] (**Table 4**). The HBM model explained 80.3% of the variance in perception of CVD preventive behavioral response over the independent variables, with the Hosmer-Lemeshow goodness of fit test for logistic regression being non-significant. Specifically, (p > 0.05).

## Discussion

The current study showed that the likelihood of CVD preventive behaviors among bank workers is 62.0%. This is lower than a study conducted in Indonesia among Ischemic Heart Disease (IHD) patients, where a relatively high level of perception of performing cardiovascular health behavior was reported in the study [25]. This might be due to the fact that people with IHD

receive advice from doctors and nurses upon admission to the hospital. However, a reduced perception of CVD risk was reported among individuals with high CVD risk [26].

Previous studies from the USA, Saudi Arabia, and others showed that understanding the respondents' perception of CVD risk protective behavior is critical for influencing behavioral change and adherence to the recommended health action in the prevention of CVD [26–28]. This study also found out that 56.6% of the respondents had good knowledge of the types of CVD, risk factors and the main preventive methods [28]. Findings from Iran and others testify to a significant correlation between CVD knowledge and CVD preventive behaviors that correspond with this study [29, 30]. The study revealed that, among HBM constructs, the perceived severity of CVD was found to be significantly associated with the likelihood of bank workers engaging in CVD preventive behaviors. A similar result was reported from Rahmati-Najarkolaei et al.'s study on students' likelihood to engage in CVD preventive behaviors [31], a study by Jorvand et al. regarding healthcare workers' CVD preventive behaviors [32], and Oruganti et al.'s study on measuring the health beliefs of hypertensive patients [33]. This indicates that respondents' belief about the severity of CVD is a key to improving their perception of CVD preventive behaviors. A previous study on employees of the Healthcare Network of Tehran [31] showed that cues to action were found to be significant predictors of respondents' engagement in CVD preventive health behaviors. The current study also corresponds with these findings.

The study revealed that there is a significant association between perceived benefit and CVD preventive behaviors. This is comparable with a study done by Kahnooji et al. concerning health workers' belief in promoting CVD preventive behavior [34], Baghianimoghadam et al.'s study on CVD preventive behavior [21], and Mohammadi et al.'s study regarding school females' CVD preventive behavior [35]. This shows that when people have a better understanding of the health benefits of the recommended actions, they are more likely to carry out CVD preventive actions. The result of this study indicated a significant association between the perceived barrier and CVD preventive behaviors, which corresponds with Sharifzadeh et al.'s study concerning the adoption of CVD preventive behaviors [29], a study conducted on CVD preventive behaviors by Khuzestan province health center employees in Iran [36], a study conducted by Baghianimoghadam et al. on CVD preventive behaviors [21], a study on Tehran University students' likelihood to engage in CVD preventive behaviors [31], and a qualitative study conducted by Sabzmakan et al. [36]. This indicates that a thorough understanding of the potential negative contributors to a specific health action, such as cost, risks and the time-consuming nature of the action, is essential to help the adoption of health behaviors by reducing the barriers. Findings of the study revealed that perceived self-efficacy is not significantly associated with bank workers' perception of CVD preventive behaviors. However, a study conducted by Sharifzadeh et al. in Iran [29], Sabzmakan et al.'s study [36], Baghianimoghadam et al.'s study [21], Rahmati-Najarkolaei et al.'s study among Tehran University students [31], Mohammadi et al.'s study among school females [35], Jordan et al.'s study [32], and with a study done by Kahnooji et al. [34] regarding CVD preventive behavior. This outlines that in various studies perceived confidence is an important factor in improving their probability of adopting health behaviors though it is not statistically significant in the current study.

The study's strength is using the tested conceptual framework, i. e HBM, which is believed to be effective in predicting the likelihood of individuals adopting the recommended health behavior and provides a theoretical basis for framing research interventions.

As a limitation, HBM is entitled to understand only cognitive aspects of an individual's perception and it is limited to addressing the individual perception. Another limitation is since the study was conducted using a cross-sectional design, it is difficult to determine whether the behavior or the predicting variable occurred first.

In conclusion, this study's findings indicated that 3 out of 5 bank workers in Hossana town have a higher perception of performing CVD preventive behaviors. The study also revealed that most of the study participants had good knowledge of CVD. Most of them were engaged in CVD preventive actions like being non-smokers, performing regular physical exercise, reducing alcohol intake, and consuming fruits and vegetables in daily meals. Moreover, bank workers exposed to passive smoking, level of alcohol consumption, regular consumption of fruits and vegetables in a daily meal, perceived severity of CVD, and cues to take preventive action were found to be predictors of their perception of engaging in CVD preventive behaviors.

## Supporting information

**S1 Questionnaire.**
(DOCX)

**S1 File.**
(DOCX)

## Acknowledgments

We would like to acknowledge the banks of Hossana town and their employees for their valuable participation in the study process. We would also like to thank the staff of Wachemo University College of Medicine and Health Sciences for their unreserved support.

## Author Contributions

**Conceptualization:** Lemlem Kifleyesus Amdemariam, Aregash Mecha Watumo, Epfrem Lejore Sibamo, Feleke Doyore Agide.

**Data curation:** Lemlem Kifleyesus Amdemariam, Aregash Mecha Watumo, Epfrem Lejore Sibamo, Feleke Doyore Agide.

**Formal analysis:** Lemlem Kifleyesus Amdemariam, Aregash Mecha Watumo, Epfrem Lejore Sibamo, Feleke Doyore Agide.

**Investigation:** Lemlem Kifleyesus Amdemariam, Feleke Doyore Agide.

**Methodology:** Lemlem Kifleyesus Amdemariam, Aregash Mecha Watumo, Epfrem Lejore Sibamo, Feleke Doyore Agide.

**Project administration:** Lemlem Kifleyesus Amdemariam, Aregash Mecha Watumo, Feleke Doyore Agide.

**Resources:** Feleke Doyore Agide.

**Software:** Aregash Mecha Watumo, Feleke Doyore Agide.

**Supervision:** Lemlem Kifleyesus Amdemariam, Aregash Mecha Watumo, Epfrem Lejore Sibamo, Feleke Doyore Agide.

**Validation:** Aregash Mecha Watumo, Epfrem Lejore Sibamo, Feleke Doyore Agide.

**Visualization:** Aregash Mecha Watumo, Feleke Doyore Agide.

**Writing – original draft:** Lemlem Kifleyesus Amdemariam, Epfrem Lejore Sibamo, Feleke Doyore Agide.

**Writing – review & editing:** Feleke Doyore Agide.

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
