## [Decision Letter · Decision Letter 0]

18 Aug 2021

PONE-D-21-13744

Perception towards Cardiovascular Diseases Preventive Practices Among Bank Workers in Hossana Town: Applying Health Belief Model

PLOS ONE

Dear Dr. Agide,

Thank you for submitting your manuscript to PLOS ONE. After careful consideration, we feel that it has merit but does not fully meet PLOS ONE’s publication criteria as it currently stands. Therefore, we invite you to submit a revised version of the manuscript that addresses the points raised during the review process.

Please address the comments appended below.

We look forward to receiving your revised manuscript.

Kind regards,

Arista Lahiri

Academic Editor

PLOS ONE

1. Please ensure that your manuscript meets PLOS ONE's style requirements, including those for file naming. The PLOS ONE style templates can be found at https://journals.plos.org/plosone/s/file?id=wjVg/PLOSOne_formatting_sample_main_body.pdf and https://journals.plos.org/plosone/s/file?id=ba62/PLOSOne_formatting_sample_title_authors_affiliations.pdf.

2. Please address the following:

- Please include additional information regarding the survey or questionnaire used in the study and ensure that you have provided sufficient details that others could replicate the analyses. For instance, if you developed a questionnaire as part of this study and it is not under a copyright more restrictive than CC-BY, please include a copy, in both the original language and English, as Supporting Information.

- Please ensure you have discussed the limitations of this study within the Discussion section, including any potential bias introduced during data collection.

Additional Editor Comments (if provided):

The authors have presented findings of their study on CVD related practices applying HBM. The authors need to address the reviewers' comments. At this moment I have one more suggestion to the authors. Please consider revising the title of the article. Particularly the last part "... applying Health Belief Model" can be improved.

Reviewers' comments:

Reviewer's Responses to Questions

**Comments to the Author**

1. Is the manuscript technically sound, and do the data support the conclusions?

Reviewer #1: Yes

Reviewer #2: Partly

2. Has the statistical analysis been performed appropriately and rigorously? 

Reviewer #1: Yes

Reviewer #2: No

3. Have the authors made all data underlying the findings in their manuscript fully available?

Reviewer #1: Yes

Reviewer #2: No

4. Is the manuscript presented in an intelligible fashion and written in standard English?

Reviewer #1: Yes

Reviewer #2: Yes

5. Review Comments to the Author

Reviewer #1: The article needs following clarifications:

1. Figure 1: Message in circular box seems to be incomplete. Please correct.

2. “since there is no study to estimate k, it is taken as 0.25” – please clarify for better understanding.

3. Figure 2: Location of PPS heading is inappropriate. It seems that only for the banks mentioned in the middle part were involved in PS allocation technique.

4. In table 2 title: September 2020 is written. Please correct.

5. Table 4: History of CVD: 0.14 (0.04, 1.07) ** - how can it be significant, since it is covering 1?

6. Most of the references are not written uniformly and as per norm. Please correct.

7. Output of binary logistic regression is missing. What about model fitness, statistical significance of model, variation of dependent variables that can be explained from the independent variables?

8. How independent variables were chosen in multivariable regression?

9. Better to reframe the title of the tables and figures. What is the necessity of providing year and month?

Reviewer #2: 1. Explanation of Health Belief Model was not mentioned in the introduction part. The title consists of "Health Belief Model". So it is better to include description of Health Belief Model in introduction part.

2. What is the rationality of assuming likelihood of taking preventive measures by only considering perceived benefit minus perceived barriers?

3. Why a total Cronbach's alpha calculated and mentioned? Already construct wise Cronbach's value was mentioned.

4. SD value of perceived severity in table 3 should be in two decimal.

5. Whether perceived self efficacy and perceived susceptibility taken into consideration as independent predictors? It should be mentioned.

6. What is the rationality of taking P-value <0.20?

7. In Discussion section, it is mentioned association between perceived benefits, perceived barriers and perceived self efficacy with preventive practices of CVD. But nowhere in the result section, these were mentioned. Explain?

8. Some references are too old. Use recent references.

6. PLOS authors have the option to publish the peer review history of their article (what does this mean?). If published, this will include your full peer review and any attached files.

Reviewer #1: **Yes: **Indranil Saha

Reviewer #2: No

---

## [Author Response · Author response to Decision Letter 0]

1 Sep 2021

Dear Academic Editor and Reviewers, 

Thank you so much for your valuable comments and interest in the publication of the manuscript. Your comments have improved the quality of our manuscript and almost all the comments are incorporated into the current manuscript. The “Revised Manuscript with Track Changes” will show how much we improved our manuscript for grammar.

Response to Dear Academic Editor, 

Thank you so much for your valuable comments and interest in the publication of the manuscript. Your comments regarding the naming of each file and submitting them with the required name are incorporated into the current manuscript. Guidelines for resubmitting each file are considered as per your comments. The title is improved as “Perception towards Cardiovascular Diseases Preventive Practices among Bank Workers in Hossana Town using the Health Belief Model.” In general, each and every comments of academic editor is considered in this current manuscript to improve the quality of the paper. We also thank the academic editor for this valuable comments and refining the manuscript more.

Response to Reviewer 1, 

Thank you so much for your insightful comments and interest in the manuscript's publication. Your suggestions improved the quality of our manuscript, and all of them have been incorporated into the current version. The points on which you require clarification were addressed one by one.

Comment 1. Figure 1: Message in circular box seems to be incomplete. Please correct.

Answer: Thank you for the comment. We improved and corrected in the current manuscript.

Comment 2. “Since there is no study to estimate k, it is taken as 0.25” – please clarify for better understanding.

Answer: We thank you for your valuable comment and clarified as it is not necessary to state in such a way. It is omitted and rephrased in better way. 

Comment 3. Figure 2: Location of PPS heading is inappropriate. It seems that only for the banks mentioned in the middle part were involved in PS allocation technique.

Answer: We thank you for the comment and amended (it was editorial error) in the current manuscript. Following comment 9, we deleted month and year in the current manuscript.

Comment 4. In table 2 title: September 2020 is written. Please correct.

Answer: We accepted your valuable comment and edited it in the current manuscript.

Comment 5. Table 4: History of CVD: 0.14 (0.04, 1.07) ** be significant, since it is covering 1?

Answer: Thank you for your valuable comment. Sorry, “history of CVD” is not significant in adjusted OR. We accepted and edited it in the current manuscript.

Comment 6. Most of the references are not written uniformly and as per norm. Please correct.

Answer: Thank you for your valuable comment. We corrected and updated in the current manuscript.

Comment 7. Output of binary logistic regression is missing. What about model fitness, statistical significance of model, variation of dependent variables that can be explained from the independent variables?

Answer: We accepted your valuable comment and included it in the current manuscript. Those variables significant in both binary and multivariable logistic regression were included in table 4. The model fitness was measured by the Hosmer-Lemeshow goodness of fit test for logistic regression. Totally, 80.3% of the variance in the perception of CVD preventive behavioral response was explained over the independent variables by the HBM model. The Hosmer-Lemeshow goodness of fit test for logistic regression was not significant. i.e. (p>0.05)

Comment 8. How independent variables were chosen in multivariable regression?

Answer: Thank you for your valuable comment. We clarified this in methods and materials part of the current manuscript. i.e. All explanatory variables that were associated with the outcome variable in bivariate analysis with a p-value of 0.20 or less were included in the initial logistic models to increase the candidate variables for final model prediction (According to Vittinghoff et al (2005, p.134)).

Comment 9. Better to reframe the title of the tables and figures. What is the necessity of providing year and month?

Answer: We rephrased and deleted month and year in the current manuscript.

Response to Reviewer #2, 

We thank you so much for your valuable comments and interest in the publication of the manuscript. We improved our manuscript as per your comment and the language is also improved accordingly. Indeed, your comments improved the quality of our manuscript and almost all the comments are incorporated into the current manuscript. The “Revised Manuscript with Track Changes” will show how much we improved our manuscript for grammar. Points that you need clarifications were answered one by one. 

Comment 1. Explanation of Health Belief Model was not mentioned in the introduction part. The title consists of "Health Belief Model". So it is better to include description of Health Belief Model in introduction part.

Answer: Thank you for your valuable comment. We accepted your comment and included the description of Health Belief Model in introduction part in the current manuscript.

Comment 2. What is the rationality of assuming likelihood of taking preventive measures by only considering perceived benefit minus perceived barriers?

Answer: Thank you for your valuable comments. The very assumption of the Health Belief Model is that people are largely rational in their thoughts and actions, and will take the best health-supporting action if they feel that it is possible to address a negative health issue. In this regard, the model not only helps to explain health-related behavior and its determinants, but it can also guide the development of interventions to influence and change health-related behavior and ultimately improve health. Actually, the risk of susceptibility and severity of the diseases lead the individual to seek a solution or to rationalize the condition as parallel. Speech on the ground (rationale), HBM based intervention is always relying on improving benefits and reducing barriers. This is called discriminative score (critical value), where the mean score of the barrier is subtracted from the mean score benefits. It is suggestive that the likelihood of taking action has a cumulative effect that ends with improving benefits and reducing barriers. I hope your previous comments to include the description of the health belief model, is very important to understand the logic of using discriminative scores to determine the likelihood of taking action.

Comment 3. Why a total Cronbach's alpha calculated and mentioned? Already construct wise Cronbach's value was mentioned.

Answer: Thank you for your valuable comment. The total Cronbach's alpha was calculated as a measure of internal consistency, that is, how closely related a set of items are as a group. It is considered to be a measure of scale reliability. In fact, construct wise calculation is enough as per your comment, and a high value for alpha does not imply that the measure is one-dimensional. We accepted and amended it in the current manuscript.

Comment 4. SD value of perceived severity in table 3 should be in two decimal.

Answer: We accepted your valuable comment and edited as two decimal places in the current manuscript.

Comment 5. Whether perceived self-efficacy and perceived susceptibility taken into consideration as independent predictors? It should be mentioned.

Answer: Thank you for your valuable comment. At the very beginning, Rosenstock's Health Belief Model (HBM) is a theoretical model concerned with health decision-making. The model attempts to explain the conditions under which a person will engage in individual health behaviors such as preventative screenings or seeking treatment for a health condition if he/she thinks as susceptible and its severity harms him/her. The confidence he has will also determine whether to engage in healthy behavior. Accepting a certain behavior as a preventive action depends on once susceptibility, severity and self-efficacy (Rosenstock, 1966). We accepted and clarified it in the introduction part of the current manuscript.

Comment 6. What is the rationality of taking P-value <0.20?

Answer: Thank you for the comment. According to The only reason to include all explanatory variables that were associated with the outcome variable in bivariate analysis with a p-value of 0.20 or less were included in the initial logistic models is to increase the candidate variables for final model prediction and to remove confiders ((According to Vittinghoff et al (2005, p.134)). And also Maldonado and Greenland (1993) suggest that potential confounders be eliminated only if p => 0.20, in order to protect against residual confounding. To briefly summarize, a crude odds ratio is just an odds ratio of one independent variable for predicting the dependent variable in the presence of confiders. The adjusted odds ratio holds other relevant variables constant and provides the odds ratio for the potential variable of interest which is adjusted for the other independent variables included in the model. We accepted and clarified it in such a way in the current manuscript. 

Comment 7. In Discussion section, it is mentioned association between perceived benefits, perceived barriers and perceived self-efficacy with preventive practices of CVD. But nowhere in the result section, these were mentioned. Explain?

Answer: Thank you for your valuable comment. Self-efficacy is not significant variable. Unfortunately, the word “not” is missed. Now, it is corrected and written in meaningful way. Concerning perceived benefits and perceived barriers, when we saw compositely, both are treated as outcome variable (your previous comment to add the description of health belief model gives a clue for this. Thank you for that). However, when treated separately with CVD prevention activities, they are statistically significant. It is quite important to discuss the two perception variables since they are treated as an outcome variable and we didn’t put their odds ratio in the previous document. However, we included in the table 4 as per your comment. We really appreciate your in-depth looking of our manuscript to increase the quality of it. We fully accepted and edited in the current manuscript.

Comment 8. Some references are too old. Use recent references.

Answer: Thank you for your valuable comment. We updated them in the current manuscript. In fact, some of the references are general truth about HBM and its application and also the trend of the occurrences of the diseases might be there.

---

## [Decision Letter · Decision Letter 1]

4 Oct 2021

PONE-D-21-13744R1Perception towards Cardiovascular Diseases Preventive Practices among Bank Workers in Hossana Town using the Health Belief ModelPLOS ONE

Dear Dr. Agide,

Thank you for submitting your manuscript to PLOS ONE. After careful consideration, we feel that it has merit but does not fully meet PLOS ONE’s publication criteria as it currently stands. Therefore, we invite you to submit a revised version of the manuscript that addresses the points raised during the review process.

Please address reviewer 1's comments.

We look forward to receiving your revised manuscript.

Kind regards,

Arista Lahiri

Academic Editor

PLOS ONE

Journal Requirements:

Additional Editor Comments:

The authors have improved the article incorporating the revisions. They, however, still need to address the comments by reviewer 1.

Reviewers' comments:

Reviewer's Responses to Questions

**Comments to the Author**

1. If the authors have adequately addressed your comments raised in a previous round of review and you feel that this manuscript is now acceptable for publication, you may indicate that here to bypass the “Comments to the Author” section, enter your conflict of interest statement in the “Confidential to Editor” section, and submit your "Accept" recommendation.

Reviewer #1: (No Response)

Reviewer #2: All comments have been addressed

2. Is the manuscript technically sound, and do the data support the conclusions?

Reviewer #1: Yes

Reviewer #2: Yes

3. Has the statistical analysis been performed appropriately and rigorously? 

Reviewer #1: Yes

Reviewer #2: Yes

4. Have the authors made all data underlying the findings in their manuscript fully available?

Reviewer #1: Yes

Reviewer #2: Yes

5. Is the manuscript presented in an intelligible fashion and written in standard English?

Reviewer #1: Yes

Reviewer #2: Yes

6. Review Comments to the Author

Reviewer #1: The article still needs following clarifications:

1. In abstract study period was written as 11 to 20th February, 2020, while in main text, method section it is written as 11 to 30th February, 2020.

2. Materials and methods: “Study design and populations: A cross-sectional study was conducted to assess bank workers’ perception of CVD preventive behaviors in Hossana Town, Ethiopia.” – need not to mention in this section.

3. Table 3: The scores were represented in mean and SD. Did you check for distribution of the scores?

4. References are still not uniform: Reference: 13 & 14 etc. (et al after name of first author); reference 20 & 21 etc.: name of 2 or 3 authors are written.

Reviewer #2: (No Response)

7. PLOS authors have the option to publish the peer review history of their article (what does this mean?). If published, this will include your full peer review and any attached files.

Reviewer #1: **Yes: **Indranil Saha

Reviewer #2: No

---

## [Author Response · Author response to Decision Letter 1]

1 Dec 2021

Dear Academic Editor and Reviewers, 

Thank you so much for your valuable comments and interest in the publication of the manuscript. Your comments have improved the quality of our manuscript and almost all the comments are incorporated into the current manuscript. The "Revised Manuscript with Track Changes" will show how much we improved our manuscript for grammar.

Response to Dear Academic Editor, 

Thank you so much for your interest in the publication of the manuscript. We revised the comments of reviewer 1. We used EndNote to revise the reviewer 1 comment about references. We will also revise at the time of proofreading/author proof, if any.

Response to Reviewer 1, 

We thank you so much for your comments and refining our manuscript for publication. Your comments were incorporated into the current version of the revised manuscript. Points that you need clarification on were answered one by one.

Comment 1. In the abstract, the study period was written as 11 to 20th February, 2020, while in the main text, the method section; it was written as 11 to 30th February, 2020.

Answer: Thank you for the comment. We improved and corrected the study period as of 11 to 30th February, 2020 in both sections in the current manuscript. Sorry, this mistake occurred when we were summarizing the manuscript. Now it is corrected.

Comment 2. Materials and methods: "study design and populations: A cross-sectional study was conducted to assess bank workers’ perception of CVD preventive behaviors in Hossana town, Ethiopia. Need not to mention in this section.

Answer: We incorporated your valuable comment in the current manuscript. It is omitted and rephrased in a better way. In our opinion, mentioning study design is very important to better understanding the description and arrangement of the study. This in turn helps to guide analysis and to answer research questions.

Comment 3. Table 3: scores were represented in mean and SD. Did you check for distribution of the scores.

Answer: Yes, dear reviewer. We thank you for the comment. We checked all the assumptions, whether they met or not.

Comment 4. References are still not uniform. References: 13 &14 etc. (et alafter name of first author); references 20 & 21 etc: 2 or 3 authors are written.

Answer: Thank you for your valuable comment. We updated them in the current manuscript and used "EndNote" to revise references as per your comment and replace them where necessary. Now it is uniform. 

Thank you again.

---

## [Decision Letter · Decision Letter 2]

4 Feb 2022

Perception towards Cardiovascular Diseases Preventive Practices among Bank Workers in Hossana Town using the Health Belief Model

PONE-D-21-13744R2

Dear Dr. Agide,

We’re pleased to inform you that your manuscript has been judged scientifically suitable for publication and will be formally accepted for publication once it meets all outstanding technical requirements.

Kind regards,

Arista Lahiri

Academic Editor

PLOS ONE

Additional Editor Comments (optional):

Reviewers' comments:

Reviewer's Responses to Questions

**Comments to the Author**

1. If the authors have adequately addressed your comments raised in a previous round of review and you feel that this manuscript is now acceptable for publication, you may indicate that here to bypass the “Comments to the Author” section, enter your conflict of interest statement in the “Confidential to Editor” section, and submit your "Accept" recommendation.

Reviewer #1: All comments have been addressed

Reviewer #2: All comments have been addressed

2. Is the manuscript technically sound, and do the data support the conclusions?

Reviewer #1: Yes

Reviewer #2: Yes

3. Has the statistical analysis been performed appropriately and rigorously? 

Reviewer #1: Yes

Reviewer #2: (No Response)

4. Have the authors made all data underlying the findings in their manuscript fully available?

Reviewer #1: Yes

Reviewer #2: (No Response)

5. Is the manuscript presented in an intelligible fashion and written in standard English?

Reviewer #1: Yes

Reviewer #2: (No Response)

6. Review Comments to the Author

Reviewer #1: The article is suitable to be published in PLOS One. Authors have addressed all my previous queries satisfactorily.

Reviewer #2: (No Response)

7. PLOS authors have the option to publish the peer review history of their article (what does this mean?). If published, this will include your full peer review and any attached files.

Reviewer #1: **Yes: **Indranil Saha

Reviewer #2: **Yes: **Sweety Suman Jha

---

## [Editor Report · Acceptance letter]

18 Feb 2022

PONE-D-21-13744R2 

Perception towards Cardiovascular Diseases Preventive Practices among Bank Workers in Hossana Town using the Health Belief Model 

Dear Dr. Agide:

I'm pleased to inform you that your manuscript has been deemed suitable for publication in PLOS ONE. Congratulations! Your manuscript is now with our production department. 

Kind regards, 

on behalf of

Dr. Arista Lahiri 

Academic Editor

PLOS ONE